# Learning to Plan & Reason for Evaluation with Thinking-LLM-as-a-Judge

**Swarnadeep Saha** [1]  **Xian Li** [1]  **Marjan Ghazvininejad** [1]  **Jason Weston** [1]  **Tianlu Wang** [1]

## Abstract

LLM-as-a-Judge models generate chain-of-thought (CoT) sequences intended to capture the step-by-step reasoning process that underlies the final evaluation of a response. However, due to the lack of human-annotated CoTs for evaluation, the required components and structure of effective reasoning traces remain understudied. Consequently, previous approaches often (1) constrain reasoning traces to hand-designed components, such as a list of criteria, reference answers, or verification questions and (2) structure them such that planning is intertwined with the reasoning for evaluation. In this work, we propose EvalPlanner, a preference optimization algorithm for Thinking-LLM-as-a-Judge that first generates an unconstrained evaluation plan, followed by its execution, and then the final judgment. In a self-training loop, EvalPlanner iteratively optimizes over synthetically constructed evaluation plans and executions, leading to better final verdicts. Our method achieves a new state-of-the-art performance for generative reward models on RewardBench and PPE, despite being trained on fewer amount of, and synthetically generated, preference pairs. Additional experiments on other benchmarks like RM-Bench, JudgeBench, and FollowBenchEval further highlight the utility of both planning and reasoning for building robust LLM-as-a-Judge reasoning models.

## 1. Introduction

As large language models (LLMs) continue to improve, reliably evaluating their long-form outputs has become even more challenging. Owing to the high cost of human evaluation, the LLM-as-a-Judge paradigm has emerged as a promising alternative where LLMs themselves are employed as evaluators (Zheng et al., 2023; Kim et al., 2024a; Saha et al., 2024a; Dubois et al., 2024). LLM-as-a-Judge models also serve as reward models during training for iterative preference optimization and self-improvement (Yuan et al., 2024). Compared to traditional reward models that only output scalar scores, LLM-as-a-Judge models expend more test-time compute by generating Chain-of-Thought (CoT) rationales of the underlying reasoning process of evaluation. This has been shown to not only improve evaluation accuracy but also enhance transparency (Zheng et al., 2023; Wang et al., 2024c; Ankner et al., 2024).

Despite the promise of LLM-as-a-Judge models, the lack of human-annotated CoTs makes it difficult to train such models. Hence, a crucial step in building these judges is generating rationales by writing down detailed evaluation instructions or rubrics that LLMs can follow. These hand-crafted instructions vary for every new domain (e.g., safety versus coding) (Yu et al., 2024b) and include manually designing evaluation criteria (Zheng et al., 2023; Saha et al., 2024a; Trivedi et al., 2024; Wang et al., 2024b;c), scoring rubrics, and steps for each criterion (Yuan et al., 2024; Trivedi et al., 2024; Kim et al., 2024b; Wang et al., 2024d). This is limiting because different tasks necessitate evaluation standards or procedures tailored to each specific task. For instance, evaluating an essay requires measuring quality along multiple, potentially subjective, fine-grained criteria like relevance and clarity whereas evaluating a math problem requires objectively verifying the correctness of the solution in a step-by-step manner (Lightman et al., 2024). Simply using predefined evaluation prompts hurts evaluation accuracy, while manually adjusting the evaluation instructions is neither scalable nor realistic, given the wide range of arbitrary and complex tasks that LLM-as-a-Judge models are used for.

To overcome these limitations, we propose EvalPlanner, a novel approach to building Thinking-LLM-as-a-Judge models that teaches LLMs to both plan and reason for evaluation. EvalPlanner is trained to perform complex evaluation by thinking and spending more test-time compute with CoTs that are decoupled into a planning component and a reasoning component. In the planning component, the model generates a detailed *evaluation plan*, that consists of all the necessary steps to evaluate responses *specific* to the given instruction. In the reasoning component, the model *executes the plan* step-by-step and reasons through the input

[1]FAIR at Meta. Correspondence to: Swarnadeep Saha <swarnadeep@meta.com>.

*Proceedings of the 42nd International Conference on Machine Learning*, Vancouver, Canada. PMLR 267, 2025. Copyright 2025 by the author(s).

---

**Example Input and Output of EvalPlanner**

**User Instruction:** [Text]There is a French pub called The Phoenix that has a low customer rating but is child friendly and is more than 30 dollars and is located in the riverside area near Crowne Plaza Hotel.
[Question]According to the above [Text], extract the attributes (name, eatType) of the restaurant.
Do not output your analysis, only output the result in the format of "attribute1[value1], attribute2[value2], ..."

**Response A:** Name[The Phoenix], eatType[pub], location[Riverside area near Crowne Plaza Hotel], priceRange[More than 30 dollars]

**Response B:** Name[The Phoenix], eatType[pub]

**Output:**
[Start of Evaluation Plan]
To evaluate the response quality, we will follow these steps:
1. **Check if the response provides the correct attributes**: Verify that the response includes the attributes "name" and "eatType" as requested in the question.
2. **Verify the accuracy of the attribute values**: Check if the values provided for the attributes match the information given in the original text.
3. **Check for any additional or missing information**: Ensure that the response does not include any extra attributes or information not requested in the question.

Step-by-step reference answer:
1. The response should include the attributes "name" and "eatType".
2. The values for the attributes should be:
        * name: The Phoenix
        * eatType: French pub
3. The response should not include any additional attributes or information.
[End of Evaluation Plan]

[Start of Plan Execution]
**Step 1: Check if the response provides the correct attributes**
Assistant A's response includes the attributes "Name", "eatType", "location", and "priceRange". However, the question only requests the attributes "nameänd "eatType". Assistant B's response includes the attributes "name" and "eatType", which matches the request.

**Step 2: Verify the accuracy of the attribute values**
...

**Step 3: Check for any additional or missing information**
Assistant A's response includes additional attributes "location" and "priceRange" that are not requested in the question. Assistant B's response does not include any additional attributes or information.

Based on the evaluation, both responses have some inaccuracies. However, Assistant B's response is more accurate in terms of providing only the requested attributes, even though it is missing the "French" descriptor. The final verdict is: [[B]]
[End of Plan Execution]

[Final Verdict]
[[B]]

---

*Figure 1.* A representative input and (truncated) output of EvalPlanner. EvalPlanner takes a user instruction and a pair of responses as inputs. It generates a Chain-of-Thought, structured into a planning component (the evaluation plan), a reasoning component (the plan execution), and the final verdict. The evaluation plan specifies the recipe while the plan execution follows this recipe step-by-step by analyzing the responses, leading to the final judgment.

response(s) to arrive at the final verdict. EvalPlanner is iteratively trained in a self-improving loop (Yuan et al., 2024; Wang et al., 2024c; Wu et al., 2024a) by sampling multiple plans and plan executions from the current model and performing preference optimization over correct and incorrect CoTs, i.e., chosen and rejected (plan, execution, verdict) triples. This teaches the model to iteratively optimize for both (1) generating a good plan that may encapsulate the most relevant and fine-grained criteria, scoring rubrics, reference answers, unit tests, etc based on the input task at hand and (2) performing correct execution grounded in the generated plan. EvalPlanner achieves this learning using only synthetic data as supervision via self-training.

We conduct extensive experiments on five reward modeling benchmarks – RewardBench, PPE, RM-Bench, JudgeBench, and FollowBenchEval – spanning instructions across categories of Chat, Safety, Code, Math, and fine-grained multi-level constraints. On RewardBench and PPE, EvalPlanner achieves new state-of-the-art scores (e.g., 93.9 on Reward-Bench) for generative reward models, outperforming base-

lines that train on up to 30x more, and typically human-annotated, data. Our model also generalizes well to other benchmarks, obtaining up to 13% improvement over a leading model for complex prompts that require evaluating multi-level constraint satisfaction. Finally, we conduct a set of comprehensive ablations that highlight the effectiveness of EvalPlanner's (1) unconstrained evaluation plans over constrained ones, (2) iterative optimization recipe of these plans, and (3) data-efficiency, allowing it to obtain competitive performance with as few as 5K synthetic preference pairs. Overall, EvalPlanner opens up new opportunities for building Thinking-LLM-as-a-Judge models that scale up test-time compute for robust and transparent evaluation by learning to both plan and reason jointly.

## 2. EvalPlanner

We consider the setting of pairwise response evaluation using the LLM-as-a-Judge paradigm (Zheng et al., 2023). The judge model takes an instruction $x$ and a pair of responses $a$ and $b$ as inputs and generates a preference judgment $y$, predicting the better response, $a$ *or* $b$. By doing so, the model also generates a Chain-of-Thought (CoT) (Wei et al., 2022) aiming to capture the step-by-step reasoning behind the evaluation process.

### 2.1. Method Overview

Evaluating long machine-generated responses to complex instructions is primarily a planning and reasoning problem. In particular, the evaluator must first *plan* the evaluation recipe and then *reason* through that recipe and the response(s) to arrive at the final verdict. With that motivation, EvalPlanner hypothesizes that an effective Chain-of-Thought for evaluation should consist of three components: (1) *the Evaluation Plan $z$*, (2) *the Execution of the Plan $e$*, and (3) the *Final Verdict $y$*. Figure 1 shows an example highlighting these three components. For a given input instruction $x$, the evaluation plan specifies the recipe for evaluating given responses to the instruction. The execution of the plan is responsible for actually conducting the evaluation by following the plan step-by-step, analyzing the input pair of responses $a$ and $b$ and generating the final judgment $y$. Given an LLM operating as an LLM-as-a-Judge, parameterized by $\theta$, where the plan $z$ and the execution $e$ are assumed to be latent variables, we can write the generative process of the final verdict $y$ as follows.

$$p_\theta(y|x,a,b) = \sum_{z \in \mathcal{P}} \sum_{e \in \mathcal{E}} p_\theta(y|e,z,x,a,b)p_\theta(e|z,x,a,b)p_\theta(z|x)$$

We follow this generative process to build preference pairs of CoTs (Section 2.2) for training such a model. See Figure 2 for an overview. Given an instruction and a seed model, we first sample multiple plans $z \in \mathcal{P}$. Then, for a given plan, instruction, and a pair of responses, we sample multiple executions $e \in \mathcal{E}$ of the plan which either lead to

the correct final verdict or not. Using this data, we develop a self-training loop that trains an LLM-as-a-Judge model by optimizing over both plans and executions, leading to better judgments (Section 2.3). At test time, the model generates CoTs of the form $\tilde{y} = (\tilde{z}, \tilde{e}, \tilde{y})$, structured into a plan, its execution, and the final verdict.

### 2.2. Synthetic Training Data Generation

LLM-as-a-Judge models are typically trained on human-annotated preference judgments. However, collecting such data is a costly and tedious process, often requiring expert annotations for domains like code and mathematics (Ouyang et al., 2022; Wang et al., 2024c). Even when such judgments exist, they do not come with any corresponding reasoning steps. This motivates us to develop EvalPlanner by only assuming access to some carefully-chosen input instructions as training data. In the rest of this section, we describe our synthetic training data generation process, which includes constructing both preference pairs $(a, b)$ and their CoTs **y**.

**Prompt Selection and Generating Response Pairs.** We choose prompts belonging to general instruction-following as well as mathematical reasoning. For general instruction-following prompts, we use the same approach as in Self-Taught Evaluators (Wang et al., 2024c) to generate response pairs, i.e., by first modifying the original instruction into a 'noisy' instruction and then generating a response to the noisy instruction. Consequently, the response to the original instruction becomes the chosen response, while the one for the 'noisy' instruction becomes the rejected response. For prompts specific to math reasoning, we sample multiple responses, where responses that lead to the correct solutions become our chosen responses, while those with incorrect solutions are considered rejected responses.

**Generating Evaluation Plans.** Given these synthetic preference pairs, we now want to generate the latent evaluation plans. Intuitively, a plan that evaluates an open-ended writing question would be structurally and semantically very different from a plan that evaluates a coding question. Hence, depending on the evaluation domain, the plans could vary significantly (see Appendix C for some examples of diverse plans). This makes manually defining the structure or the components of a good plan time-consuming, less generalizable, and prone to user biases. Thus, we design a generic and unconstrained *plan generation* prompt (Fig. 3 in Appendix) that queries a seed model (e.g., an instruction-tuned LLM) for an *initial* plan conditioned only on the input instruction. These plans will then be optimized later by self-training. As part of our experiments, we also show the efficacy of this plan generation prompt against other prompts that try to constrain plans to certain pre-defined components. Note that our planning prompt does not condition on the response pair to ensure that the generated plans represent only the

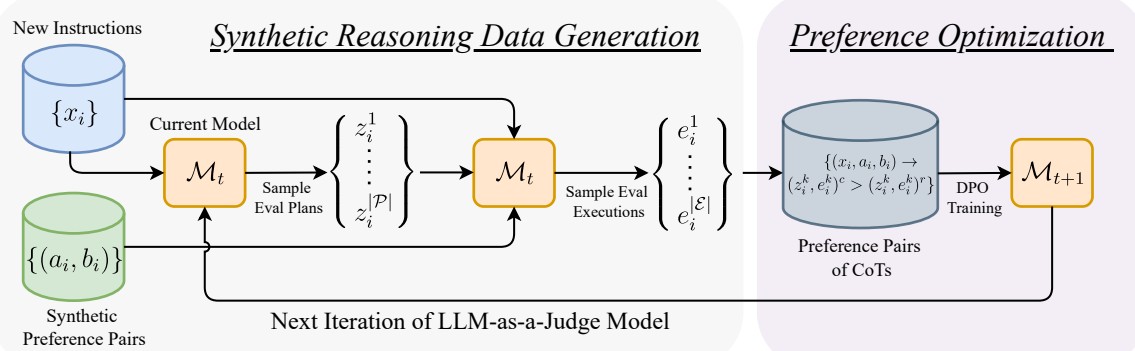

*Figure 2.* **EvalPlanner:** A Thinking-LLM-as-a-Judge model that learns to think by planning and reasoning for evaluation. Given an instruction and a preference pair as input, the synthetic reasoning data generation recipe consists of sampling multiple plans and multiple executions from the current model. These evaluation plans and executions are used to construct preference pairs of Chain-of-Thoughts, which are then iteratively optimized with DPO in a self-training loop.

recipe and not the actual evaluation. Evaluation happens in the second stage of plan execution, as described below.

**Generating Plan Executions.** In this second stage of *plan execution*, we now prompt the same seed model with the instruction and the response pair to *reason* through the plan generated in the previous step and the response pairs to produce a verdict (Fig. 4 in Appendix). The benefits of disentangling the planning and execution stages are two-fold. First, the disentanglement tries to enforce that the reasoning/execution follows the plan. Second, by sampling multiple plans and multiple executions for the same plan, we can train a model on diverse evaluation data that vary in both planning and execution. Like the initial plans, the initial plan executions will also be optimized later via self-training.

**Building Preference Pairs of Plans & Executions.** Given the preference pairs, the plans, and their executions, we now build a preference tuning dataset to optimize over the CoTs. For each input instruction, we sample $|\mathcal{P}|$ plans and for each plan, we sample $|\mathcal{E}|$ executions. To account for position bias (Zheng et al., 2023), we generate plan executions according to both orders of the response pairs – $(a, b)$ and $(b, a)$. This results in a total of $2 \times |\mathcal{P}| \times |\mathcal{E}|$ CoTs, for each input instruction. A thought is considered correct if the corresponding (plan, execution, judgment) triplet leads to the correct verdict and incorrect otherwise. Using this correctness criterion, we construct our preference tuning dataset $\mathcal{D} = \mathcal{D}^{(a,b)} \cup \mathcal{D}^{(b,a)}$ where $\mathcal{D}^{(a,b)}$ is defined as follows and $\mathcal{D}^{(b,a)}$ swaps the execution order of responses.

$$\mathcal{D}^{(a,b)} = \bigcup_{z \in \mathcal{P}} \{(x, a, b) \to (z, e^c, y^c); (x, a, b) \to (z, e^r, y^r)$$
$$|e^c, e^r \in \mathcal{E}^{(a,b)}\}$$

Intuitively, given an input $(x, a, b)$, we generate multiple executions for each sampled plan $z$ to enable the model to learn from its executions. Specifically, for each plan, we generate multiple executions and construct all possi-

ble correct and incorrect execution-pairs leading to correct and incorrect judgments respectively. This process allows us to construct "chosen" (c) and "rejected" (r) pairs that differ in their executions. To further improve the model's performance, we repeat this construction process. We construct chosen and rejected pairs for all plans $\mathcal{P}$, enabling the model to learn to generate better plans in the first place. We also construct pairs for both execution orders of responses – $(a, b)$ and $(b, a)$ – ensuring that the model becomes position-consistent. If none of the executions lead to the correct verdict, the corresponding sample is not included in our training data. By scaling up the construction process, we provide the model with a more comprehensive set of examples to learn from, enabling it to refine its decision-making process and improve its overall performance.

### 2.3. Preference Optimization of Plans & Executions

Having developed the initial training data generation recipe, we now describe the training algorithm of EvalPlanner. The pipeline consists of a self-training loop, starting with a seed model $\mathcal{M}_0$ (e.g., an instruction-tuned LLM), doing supervised fine-tuning (SFT) on a subset of the 'chosen' CoTs to obtain a model $\mathcal{M}_1^{\text{SFT}}$, followed by two iterations of Direct Preference Optimization (DPO) (Rafailov et al., 2024) on preference pairs of CoTs, leading to models $\mathcal{M}_1^{\text{DPO}}$ and $\mathcal{M}_2^{\text{DPO}}$.

**$\mathcal{M}_1^{\text{SFT}}$: SFT on $\mathcal{D}_1^c$, initialized from $\mathcal{M}_0$.** Starting from the seed model $\mathcal{M}_0$ and a subset of input instructions and response pairs, we follow the recipe in Section 2.2 to generate the preference pairs of thoughts. Let us denote this dataset by $\mathcal{D}_1$. To teach the model to correctly follow the pattern of our CoT (plan+execution+verdict), we first fine-tune $\mathcal{M}_0$ on $\mathcal{D}_1^c$ – a subset of only the 'chosen' thoughts from $\mathcal{D}_1$. Specifically, for each instruction, we randomly sample one correct thought (that leads to the correct verdict) and perform SFT on that data, leading to a model $\mathcal{M}_1^{\text{SFT}}$.

$\mathcal{M}_1^{\mathtt{DPO}}$: **DPO on $\mathcal{D}_1$, initialized from $\mathcal{M}_1^{\mathtt{SFT}}$.** Next, initialized from $\mathcal{M}_1^{\mathtt{SFT}}$, we perform DPO on the dataset $\mathcal{D}_1$, consisting of both chosen and rejected thoughts. Given the two distinct parts of plan and execution tokens in the thoughts, this teaches the model to contrast between correct and incorrect thoughts, that vary in both the plan and the execution of evaluation. We thus obtain a model $\mathcal{M}_1^{\mathtt{DPO}}$.

$\mathcal{M}_2^{\mathtt{DPO}}$: **DPO on $\mathcal{D}_2$, initialized from $\mathcal{M}_1^{\mathtt{DPO}}$.** EvalPlanner also consists of a second iteration of DPO, wherein we choose a fresh subset of instructions and response pairs and generate CoTs using the same recipe but from the previous iteration of model $\mathcal{M}_1^{\mathtt{DPO}}$. In particular, we first sample $|\mathcal{P}|$ CoTs from $\mathcal{M}_1^{\mathtt{DPO}}$ for each training data point, separate out the plans from the thoughts, and then use the same $\mathcal{M}_1^{\mathtt{DPO}}$ model to sample $|\mathcal{E}|$ executions for each plan. We denote this second iteration of CoT data as $\mathcal{D}_2$. We train on new inputs and thoughts from an updated model, under the assumption that the data from the previous iteration is of lower quality. Empirically, we also show that this outperforms a single iteration of DPO trained on the entire set of inputs.

## 3. Experimental Setup

**Training.** We select prompts from two different sources – WildChat (Zhao et al., 2024) and MATH (Hendrycks et al., 2021). For WildChat, we directly use the synthetic responses generated by Self-Taught Evaluators (Wang et al., 2024c). For MATH questions, we generate synthetic responses as follows. We prompt a Mixtral 22Bx8 Instruct model to generate multiple candidate solutions. The responses that lead to the correct final answers become our chosen responses while those with incorrect final answers are considered rejected responses. Using synthetic response-pair generation, we collect a total of $17,588$ and $4,141$ unique (instruction, chosen, rejected) triples from WildChat and MATH, respectively, as our training data, using two separate methods. From this, we select a random subset of 5K instructions (consisting of 2.5K from WildChat and 2.5K from MATH) for SFT and the first iteration of DPO. We reserve the rest for the second iteration of DPO. In each iteration, we sample 5 plans and for each plan, we sample 8 executions (4 in each order of response pair) using a temperature of $0.8$ and top_p of $0.95$. We develop EvalPlanner with either Llama-3.1-70B-Instruct or Llama-3.3-70B-Instruct as the seed model to show the generalizability of our approach across multiple seed models. As validation set, we choose 150 samples from each of WildChat and MATH, which we use for checkpoint selection. To account for position bias in pairwise evaluation, we double the number of examples in the validation set by considering both orders of response pairs. We use the fairseq2 library (Balioglu, 2023) for model training and vLLM (Kwon et al., 2023) for inference. All models are trained for a maximum of 1K steps, saving check-

points every 100 steps and doing early stopping based on the validation set. Detailed training hyperparameters are provided in Table 12.

**Evaluation.** We test EvalPlanner on the following pairwise evaluation benchmarks.

- **RewardBench** (Lambert et al., 2024). It consists of (prompt, chosen, rejected) triples spanning 4 categories of prompts: chat, chat-hard, safety, and reasoning.
- **Preference Proxy Evaluations (PPE)** (Frick et al., 2025). PPE is a large-scale benchmark that links reward models to real-world human preference performance. It consists of two subsets: (i) **PPE Preference** (10.2K samples), human preference pairs from Chatbot Arena featuring 20 LLMs in 121+ languages, and (ii) **PPE Correctness** (12.7K samples), response pairs from four models across popular verifiable benchmarks (MMLU-Pro, MATH, GPQA, MBPP-Plus, IFEval). The first subset evaluates subjective preferences, while the second tests alignment in Best-of-N tasks.
- **FollowBenchEval**. We build this new evaluation benchmark from FollowBench (Jiang et al., 2024). The original benchmark consists of complex prompts that test LLMs' ability to follow multi-level fine-grained constraints (e.g., 'Write a summary within 20 words'). We convert this benchmark into a pairwise evaluation benchmark by sampling two responses from a single model (LLama-3.1-8B-Instruct, LLama-3.2-3B-Instruct, or Mistral-7B-Instruct-v0.2) such that one response satisfies all the constraints and the other one does not. Note that by generating the response-pair using the same model, we ensure consistency in response style which can otherwise lead to potentially superficial features for preference judgments. Our evaluation benchmark, called FollowBenchEval, comprises of 205 samples and spans five different constraint-types of Content, Situation, Style, Format, and Example. This benchmark specifically tests LLM-based judges' ability to (1) plan for multiple constraints that need to be checked, and (2) produce a verdict by checking for those constraints.
- **RM-Bench** (Liu et al., 2024). RM-Bench is designed to assess the robustness of reward models, based on their sensitivity and resistance to subtle content differences and style biases. The original benchmark primarily focuses on evaluating reward models that rate each response independently. We modify the input prompt to accommodate for the evaluation of LLM-as-a-Judge models, which conduct pairwise judgments by comparing two responses simultaneously.
- **JudgeBench** (Tan et al., 2024). JudgeBench is a recent benchmark that evaluates LLM-based judges on challenging response pairs spanning knowledge, reasoning, math, and coding. It sources input instructions

*Table 1.* Comparison of EvalPlanner with SOTA generative reward models on RewardBench. EvalPlanner outperforms all prior models, while using a smaller number of (22K) synthetically constructed preference pairs as training data. †: Results taken from either RewardBench leaderboard or the corresponding paper. ‡: Results taken from the Critic-RM-Rank paper (Yu et al., 2024b).

| Models | #Pref Pairs | Overall | Chat | Chat-Hard | Safety | Reasoning |
|---|---|---|---|---|---|---|
| *Open and Closed LLMs* | | | | | | |
| Llama3.1-70B-Instruct[†] | - | 84.0 | 97.2 | 70.2 | 82.8 | 86.0 |
| Llama3.1-405B-Instruct[†] | - | 84.1 | 97.2 | 74.6 | 77.6 | 87.1 |
| Llama3.3-70B-Instruct | - | 85.4 | 96.9 | 77.4 | 77.6 | 89.6 |
| Claude-3.5-sonnet[†] | - | 84.2 | 96.4 | 74.0 | 81.6 | 84.7 |
| GPT-4o[†] | - | 86.7 | 96.1 | 76.1 | 88.1 | 86.6 |
| Gemini-1.5-pro-0514[†] | - | 88.2 | 92.3 | 80.6 | 87.9 | 92.0 |
| *Reward Models with Critiques* | | | | | | |
| SynRM[‡] (Ye et al., 2024) | - | 87.3 | 97.5 | 76.8 | 86.3 | 88.5 |
| CLoud[‡] (Ankner et al., 2024) | - | 87.6 | 98.0 | 75.6 | 89.0 | 87.6 |
| Critic-RM-Rank[‡] (Yu et al., 2024b) | - | 90.5 | 97.5 | 79.6 | 94.1 | 90.6 |
| *SOTA Generative Reward Models* | | | | | | |
| Self-Taught Evaluator[†] (Wang et al., 2024c) | 20K | 90.0 | 96.9 | 85.1 | 89.6 | 88.4 |
| SFR-Llama-3.1-70B-Judge[†] (Wang et al., 2024b) | 680K | 92.7 | 96.9 | 84.8 | 91.6 | 97.6 |
| Skywork-Critic-Llama-3.1-70B[†] (Shiwen et al., 2024) | 80K | 93.3 | 96.6 | 87.9 | 93.1 | 95.5 |
| LMUnit[†] (Saad-Falcon et al., 2024) | 84K | 93.4 | - | - | - | - |
| **EvalPlanner (w/ Llama-3.1-70B-Instruct as seed model)** | 22K | **93.9** | 97.5 | 89.4 | 93.0 | 95.5 |
| **EvalPlanner (w/ Llama-3.3-70B-Instruct as seed model)** | 22K | 93.8 | 97.7 | 89.5 | 91.7 | 96.1 |

from existing datasets and generates candidate responses using stronger language models such as GPT-4o and Claude-3.5-Sonnet. Following Tan et al. (2024), we report results on the GPT-4o subset.

For RewardBench, PPE, and RM-Bench, we follow the original evaluation protocol of reporting accuracy over a single random ordering of paired responses. We report position-consistent accuracy for JudgeBench and FollowBenchEval to account for position bias. Specifically, a prediction is considered correct if the model consistently makes a correct judgment in both orders. We train and test all our models using the standard pair-wise judge prompt from prior work (Zheng et al., 2023), as shown in Figure 5. The maximum number of generation tokens is set to 2048 and the temperature to 0 for inference.

**Baselines.** We compare EvalPlanner with a range of models, including (1) Powerful Open-Sourced and Closed-Sourced LLMs used as judges in a zero-shot manner, (2) Reward Models with Critiques, capable of generating both scalar scores and critiques, and (3) SOTA Generative Reward Models, as listed on the RewardBench leaderboard. We focus on models that also generate rationales along with the final verdict, to compare related competing approaches.

## 4. Results

### 4.1. Experimental Results on Benchmarks

**EvalPlanner outperforms all baselines while being trained on fewer, and synthetically generated, prefer-**

**ence pairs.** Table 1 shows results on RewardBench. Using the same recipe, we train two EvalPlanner models with different Llama versions as the seed model (Llama-3.1-70B-Instruct and Llama-3.3-70B-Instruct). Both of our models outperform all baselines, achieving new state-of-the-arts for generative reward models on RewardBench. Particularly impressively, EvalPlanner achieves these results by being trained on a smaller number of preference pairs (22K), compared to most prior works. Moreover, EvalPlanner's training data only consists of synthetically generated preference pairs, unlike past works that primarily train on human-annotated preference pairs. EvalPlanner's training recipe is also equally performant on both Llama seed models, showing the usefulness of our initial training data and the generalizability of our approach. In Table 2, we compare EvalPlanner to DeepSeek-GRM (Liu et al., 2025), a SOTA generative reward model on PPE. Our method obtains significant improvements on the PPE Correctness subset, consisting of popular reasoning benchmarks, thereby showcasing its potential as a reward model for Best-of-N alignment. In general, our results highlight the utility of planning and reasoning for evaluation, not only for better final judgments but also for better grounding of the evaluation in a detailed plan.

**EvalPlanner's plans are tailored toward the specific instruction being evaluated.** We design EvalPlanner such that the generated plan represents a general evaluation recipe tailored toward the specific instruction being evaluated. This is achieved by disentangling planning from reasoning and having the model directly optimize the thoughts without any domain-specific tuning. Appendix C

*Table 2.* Results on PPE comparing EvalPlanner with state-of-the-art LLM-as-a-Judge and reward models. †: Results taken from Liu et al. (2025) and Frick et al. (2025).

| Models | #Training Pref. Pairs | PPE Overall | PPE Preference | Overall | MMLU-Pro | MATH | GPQA | MBPP-Plus | IFEval |
|---|---|---|---|---|---|---|---|---|---|
| | | | | | | PPE Correctness | | | |
| *Open and Closed LLM-as-a-Judge* | | | | | | | | | |
| Llama-3.1-8B-Instruct | – | 55.5 | 56.4 | 54.7 | 56.3 | 62.9 | 51.4 | 50.1 | 52.8 |
| GPT-4o† | – | 62.3 | 67.1 | 57.6 | – | – | – | – | – |
| Llama-3.3-70B-Instruct | – | 65.8 | 65.9 | 65.7 | 72.1 | 73.1 | 61.2 | 59.6 | 62.3 |
| *SOTA Generative Reward Models* | | | | | | | | | |
| DeepSeek-GRM-27B† | 237K | 62.2 | 64.7 | 59.8 | 64.8 | 68.8 | 55.6 | 50.1 | 59.8 |
| DeepSeek-GRM-27B (MetaRM voting@32)† | 237K | 65.2 | **67.2** | 63.2 | 68.1 | 70.0 | 56.9 | 50.8 | 70.4 |
| **EvalPlanner (w/ Llama-3.1-70B-Instruct)** | 22K | 66.9 | 65.8 | 68.0 | 77.8 | 79.2 | 58.6 | 63.5 | 60.9 |
| **EvalPlanner (w/ Llama-3.3-70B-Instruct)** | 22K | **67.9** | 65.6 | **70.2** | 78.4 | 81.7 | 64.4 | 62.2 | 64.3 |

*Table 3.* Results on RewardBench comparing EvalPlanner at 8B scale with larger LLM-as-a-Judge models.

| Model | Overall | Chat | Chat-Hard | Safety | Reasoning |
|---|---|---|---|---|---|
| Llama-3.1-8B-Instruct | 69.5 | 92.7 | 46.1 | 64.4 | 74.7 |
| Llama-3.1-70B-Instruct | 84.1 | 97.2 | 70.2 | 82.8 | 86.0 |
| Claude-3.5-Sonnet | 84.2 | 96.4 | 74.0 | 81.6 | 84.7 |
| EvalPlanner (w/ Llama-3.1-8B-Instruct) | 83.0 | 85.5 | 84.0 | 83.4 | 79.3 |

*Table 4.* EvalPlanner results on RewardBench comparing two iterations of DPO with one iteration.

| # DPO Iterations | #Pref Pairs | Accuracy |
|---|---|---|
| 1 (w/ Llama-3.1-70B-Instruct) | 5K | 92.3 |
| 1 (w/ Llama-3.1-70B-Instruct) | 22K | 92.5 |
| 2 (w/ Llama-3.1-70B-Instruct) | 22K (5K+17K) | 93.9 |

shows examples of such plans generated by EvalPlanner for diverse instructions.

**EvalPlanner at smaller scale matches the performance of larger models.** In Table 3, we compare EvalPlanner at the 8B scale with much larger models like Llama-3.1-70B-Instruct and Claude-3.5-Sonnet. We show that our training recipe is also effective at such smaller scales, allowing EvalPlanner-8B to match the performance of larger LLM-as-a-Judge models.

**EvalPlanner is data-efficient and benefits from iterative thought optimization.** Next, in Table 4, we show the performance of EvalPlanner with as few as 5K preference pairs. It obtains a score of 92.3, competitive with the best models on RewardBench. We also demonstrate the effectiveness of iterative DPO – the second iteration of DPO improves results significantly ($92.3 \rightarrow 93.9$). In contrast, the same amount of data in one single DPO iteration only leads to marginal improvements ($92.3 \rightarrow 92.5$). The iterative improvement of EvalPlanner can be attributed to training on newer data points that are augmented with CoTs from an updated model. Repeating this recipe for more iterations can potentially lead to further improvements, which we leave for future work to explore.

**EvalPlanner generalizes to evaluating multi-level constraints in FollowBenchEval.** Table 5 presents our results on FollowBenchEval. The input instructions contain up to five constraints, denoted in the table as L1-L5. Given the nature of this dataset, the preference judgments focus on objective preference criteria (i.e., whether all constraints are satisfied or not), as opposed to subjective metrics like stylistic preferences (e.g., Chat category in RewardBench). This makes evaluating such prompts more challenging for LLMs and allows us to objectively assess the utility of planning and step-wise reasoning for evaluation. In such scenarios, EvalPlanner demonstrates clear benefits over its baselines that do not explicitly plan or reason, outperforming Skywork-Critic-Llama-3.1-70B (a state-of-the-art model on RewardBench) by a significant 13%.

**EvalPlanner generalizes to RM-Bench and JudgeBench.** We show results on other recent benchmarks like RM-Bench (Liu et al., 2024) and JudgeBench (Tan et al., 2024) in Tables 6 and 7 respectively. On RM-Bench, EvalPlanner outperforms all baselines, achieving up to 8% improvement over a prior state-of-the-art Skywork-Critic-LLama-3.1-Instruct model, showing its robustness to subtle differences and style biases. Notably, while all other models exhibit a drop in accuracy on the hard subset, EvalPlanner is equally performant across all the subsets. On JudgeBench, EvalPlanner with LLama-3.3-70B-Instruct achieves comparable performance to Skywork-Critic-LLama-3.1-Instruct, while being trained on much less and synthetic preference pairs.

*Table 5.* Results on FollowBenchEval for evaluation of complex prompts with multi-level constraints. EvalPlanner significantly outperforms other approaches on this challenging task.

| Model | Overall | L1 | L2 | L3 | L4 | L5 |
|---|---|---|---|---|---|---|
| Llama-3.1-70B-Instruct | 44.4 | 51.1 | 50.0 | 35.9 | 46.2 | 42.4 |
| Llama-3.3-70B-Instruct | 52.2 | 55.3 | 61.9 | 48.7 | 53.8 | 45.5 |
| Self-Taught Evaluator (Wang et al., 2024c) | 46.8 | 53.2 | 52.4 | 51.3 | 43.6 | 36.4 |
| Skywork-Critic-Llama-3.1-70B (Shiwen et al., 2024) | 52.2 | 63.8 | 57.1 | 48.7 | 46.2 | 48.5 |
| EvalPlanner (w/ Llama-3.1-70B-Instruct) | 56.6 | 66.0 | 61.9 | 56.4 | 53.8 | 48.5 |
| EvalPlanner (w/ Llama-3.3-70B-Instruct) | **65.4** | 72.3 | 73.8 | 66.7 | 61.5 | 57.6 |

*Table 6.* Results on RM-Bench for evaluation of models' robustness to subtle content changes and style biases. EvalPlanner demonstrates superior robustness across all subsets, outperforming other methods which are more vulnerable to subtle changes, particularly in the Hard subset where responses are detailed and well-formatted.

| Model | Overall | Easy | Normal | Hard |
|---|---|---|---|---|
| Llama3.1-70B-Instruct | 64.9 | 68.9 | 62.6 | 63.3 |
| Llama3.3-70B-Instruct | 69.5 | 77.5 | 66.3 | 64.8 |
| Self-Taught Evaluator (Wang et al., 2024c) | 73.6 | 75.9 | 72.4 | 72.4 |
| Skywork-Critic-Llama-3.1-70B (Shiwen et al., 2024) | 74.1 | 76.3 | 72.9 | 73.1 |
| EvalPlanner (w/ Llama-3.1-70B-Instruct) | 80.0 | 81.7 | 77.2 | 81.1 |
| EvalPlanner (w/ Llama-3.3-70B-Instruct) | **82.1** | 81.1 | 80.8 | 84.3 |

*Table 7.* Results on JudgeBench for evaluation of models' capabilities on challenging questions spanning multiple categories. EvalPlanner with Llama-3.3-70B-Instruct achieves comparable performance to Skywork-Critic-Llama-3.1-70B and outperforms all other baselines.

| Model | Overall | Knowledge | Reasoning | Math | Coding |
|---|---|---|---|---|---|
| Llama3.1-70B-Instruct | 50.3 | 53.9 | 36.7 | 64.3 | 50.0 |
| Llama3.3-70B-Instruct | 48.6 | 50.0 | 43.9 | 55.4 | 45.2 |
| Self-Taught Evaluator (Wang et al., 2024c) | 48.3 | 52.6 | 40.8 | 57.1 | 38.1 |
| Skywork-Critic-Llama-3.1-70B (Shiwen et al., 2024) | **57.1** | 56.5 | 55.1 | 71.4 | 45.2 |
| EvalPlanner (w/ LLama-3.1-70B-Instruct) | 50.9 | 48.1 | 50.0 | 60.7 | 50.0 |
| EvalPlanner (w/ LLama-3.3-70B-Instruct) | 56.6 | 55.8 | 56.1 | 69.6 | 42.9 |

## 4.2. Ablations and Analysis

We conduct all ablations on RewardBench using an EvalPlanner checkpoint, trained on 2.5K MATH instructions using Llama-3.1-70B-Instruct as the seed model.

**Effectiveness of Thought Preference Optimization.** In Table 8, we compare EvalPlanner with (1) the seed Llama-3.1-70B-Instruct model, (2) a model trained to only predict the final verdict without any intermediate CoT, and (3) an EvalPlanner variant only SFT'ed on the "chosen" examples. The results show that preference optimization of plans & executions is particularly important, leading to significant improvements over all baselines.

**Effectiveness of Unconstrained Plans over Constrained Plans.** Recall that EvalPlanner is built with an initial planning prompt that relied on the seed model to generate unconstrained plans. In this experiment, we compare this unconstrained planning prompt with two other prompts that constrain the plans to (1) a list of criteria, similar to Self-Taught Evaluators (Wang et al., 2024c) or (2) a list of verification questions, similar to Chain-of-Verification (Dhuliawala et al., 2023). As shown in Table 9, unconstrained plans obtain superior performance by generating more detailed plans and then grounding the evaluation on that plan. A generic planning prompt that works across multiple domains showcases the generalizability of our approach.

Appendix A presents more analyses like effect of scaling up the number of plans and executions and source instructions.

## 5. Related Work

**LLM-as-a-Judge.** Human evaluation is often considered the gold standard for evaluating LLM responses to complex and open-ended instructions (Ouyang et al., 2022; Dubey et al., 2024). However, given the slow, expensive, and noisy nature of human evaluation (Clark et al., 2021; Karpinska

*Table 8.* Ablation on RewardBench showing the effectiveness of preference optimization of plans & executions.

| Model | Accuracy |
|---|---|
| Llama3.1-70B-Instruct (seed model) | 84.0 |
| Llama3.1-70B-Instruct (trained w/o thoughts) | 86.2 |
| EvalPlanner (SFT w/ thoughts) | 86.8 |
| EvalPlanner (SFT + DPO w/ thoughts) | 90.5 |

*Table 9.* Ablation on RewardBench comparing the effectiveness of different types of plans.

| Type of Plan | Accuracy |
|---|---|
| List of Criteria (Wang et al., 2024c) | 83.9 |
| Verification Questions (Dhuliawala et al., 2023) | 84.8 |
| Unconstrained (Ours) | 86.8 |

et al., 2021), automatic approaches leveraging LLMs have emerged as scalable and cost-effective alternatives (Zheng et al., 2023; Liu et al., 2023; Kim et al., 2024a; Saha et al., 2024a; Jiang et al., 2023; Zhu et al., 2023). Compared to reward models that only output scalar scores (Wang et al., 2024a;e;d), LLM-as-a-Judge evaluators are more robust and interpretable because of their ability to also generate detailed rationales (Zheng et al., 2023; Zhang et al., 2024a; Ankner et al., 2024). However, in the absence of any human-annotated reasoning traces for evaluation, past works have leveraged LLMs to generate these traces by writing custom prompts for every new domain (Yu et al., 2024b) and hand-designing the components and structure of CoTs, ranging from fine-grained criteria (Zheng et al., 2023; Saha et al., 2024a; Wang et al., 2024c; Zeng et al., 2024; Trivedi et al., 2024), scoring rubric (Yuan et al., 2024; Trivedi et al., 2024; Wu et al., 2024b), verification questions (Dhuliawala et al., 2023), natural language unit tests (Saad-Falcon et al., 2024), and reference answers (Zhang et al., 2024b). In contrast, EvalPlanner proposes a unifying perspective on evaluation by subsuming all necessary components for sound evaluation inside a *plan* and then letting the model optimize these plans and their executions in a self-training loop.

**Self-Alignment.** Reinforcement Learning from Human Feedback requires a large amount of human annotations, which can be expensive to obtain (Bai et al., 2022; Lee et al., 2024). This has led to the development of various self-alignment techniques for general instruction following (Li et al., 2024; Yuan et al., 2024; Wu et al., 2024a), reasoning (Zelikman et al., 2022; Pang et al., 2024; Gulcehre et al., 2023; Yu et al., 2024a), and evaluation (Pace et al., 2024; Wang et al., 2024c; Trivedi et al., 2024). Specifically, for evaluation, Wang et al. (2024c) construct preference pairs by adding noise to the original instructions, while Trivedi et al. (2024) uses self-rationalization and a meta-judge to train a fine-grained judge. Different from these, EvalPlanner proposes a novel self-training recipe that teaches an LLM-as-a-Judge to *think* by generating and reasoning with evaluation plans.

**Training to Think, Plan, and Reason.** EvalPlanner follows a large body of prior work on equipping LLMs with the ability to think by generating additional thought tokens before the final answer (Nye et al., 2021; Zelikman

et al., 2022; Wu et al., 2024a; Hosseini et al., 2024). Unlike methods that train on ground-truth thoughts e.g., in the domains of algorithmic reasoning, math, or planning (Nye et al., 2021; Lehnert et al., 2024; Saha et al., 2024b), EvalPlanner is bootstrapped and self-trained from synthetically generated thoughts – focusing on evaluation where objectively defining the structure and components of intermediate thoughts is challenging. Moreover, EvalPlanner's thoughts have decoupled planning and reasoning components, allowing it to optimize both at the same time.

## 6. Conclusion

We presented EvalPlanner, a novel approach for building robust and data-efficient Thinking-LLM-as-a-Judge models. Through comprehensive experiments across five benchmarks, we demonstrated the effectiveness of our method, achieving a new SOTA with significantly less, and synthetically generated, training data. To further understand the capabilities of Thinking-LLM-as-a-Judge models, future work could employ them as reward models in the RLHF pipeline.

## Impact Statement

EvalPlanner's broader goal is to advance the field of Machine Learning and in particular, evaluation, by allowing LLM-as-a-Judge models to think before producing a judgment. This has the potential to improve evaluation accuracy and transparency in various applications. EvalPlanner is trained on synthetically generated data from seed Llama models that can reflect stereotypes, biases, and other negative traits present in their pre-training data (Weidinger et al., 2021), which we do not have control over. We encourage further research and discussion on these topics to ensure that this technology is developed and deployed responsibly.

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

## A. More Analysis

**Scaling Number of Plans & Executions during Training.** We also study the effect of scaling the number of latent evaluation plans and executions in Table 10. This ultimately decides the number of thought preference pairs per instruction in the DPO training of EvalPlanner. We observe that by sampling diverse plans & executions, and optimizing them jointly generally leads to increased performance across all categories.

**Effect of Source of Input Instructions.** We train EvalPlanner by mixing instructions from WildChat and MATH. In Table 11, we show that while training on WildChat instructions help the Chat-Hard category more, reasoning performance is particularly enhanced by training on the MATH instructions.

## B. Prompts

Figure 3 shows the planning prompt for generating initial evaluation plans from the seed model. Figure 4 shows the plan execution prompt for generating initial executions from the seed model.

## C. Examples of plans generated by EvalPlanner

Figures 6, 7, and 8 show examples of diverse plans generated by EvalPlanner, based on the user instruction.

---

**Prompt Template for Generating Evaluation Plans**

We want to evaluate the quality of the responses provided by AI assistants to the user question displayed below. For that, your task is to help us build an evaluation plan that can then be executed to assess the response quality. Whenever appropriate, you can choose to also include a step-by-step reference answer as part of the evaluation plan. Enclose your evaluation plan between the tags "[Start of Evaluation Plan]" and "[End of Evaluation Plan]".

[User Question]
{instruction}

---

*Figure 3.* Prompt template for generating initial evaluation plans from the seed model, conditioned on the input instruction. Plans in successive iterations are generated from the previous iteration of the judge model.

---

**Prompt Template for Execution of Evaluation Plans**

Please act as an impartial judge and evaluate the quality of the responses provided by two AI assistants to the user question displayed below. Your evaluation should be performed by following the provided evaluation plan step-by-step. Avoid copying the plan when doing the evaluation. Please also only stick to the given plan and provide explanation of how the plan is executed to compare the two responses. Avoid any position biases and ensure that the order in which the responses were presented does not influence your decision. Do not allow the length of the responses to influence your evaluation. Do not favor certain names of the assistants. Be as objective as possible. After providing your evaluation, output your final verdict by strictly following this format: "[[A]]" if assistant A is better, "[[B]]" if assistant B is better.

[User Question]
{instruction}

[The Start of Assistant A's Answer]
{response A}
[The End of Assistant A's Answer]

[The Start of Assistant B's Answer]
{response B}
[The End of Assistant B's Answer]

[The Start of Evaluation Plan]
{evaluation plan}
[The End of Evaluation Plan]

---

*Figure 4.* Prompt template for generating initial executions of evaluation plans from the seed model, conditioned on the input instruction, a pair of responses, and an evaluation plan. Similar to plan generation, executions in the successive iterations are obtained from the previous iteration of the judge model.

*Table 10.* Ablation on RewardBench showing the effect of scaling the number of CoTs (per instruction) by sampling {3, 5} plans and {4, 8} executions.

| Training Data | Overall | Chat | Chat-Hard | Safety | Reasoning |
|---|---|---|---|---|---|
| 3 Plans & 4 Exec | 88.8 | 97.2 | 77.2 | 88.9 | 92.2 |
| 5 Plans & 4 Exec | 89.3 | 97.6 | 78.5 | 89.4 | 91.6 |
| 5 Plans & 8 Exec | 90.5 | 98.6 | 79.8 | 90.1 | 93.4 |

*Table 11.* Ablation on RewardBench showing the effect of source of prompts on the different categories. While Wildchat instructions help the Chat-hard category more, MATH prompts are more effective for the Reasoning category.

| Prompt Source | Chat | Chat-Hard | Safety | Reasoning |
|---|---|---|---|---|
| MATH (2.5K) | 98.6 | 79.8 | 90.1 | 93.4 |
| Wildchat (2.5K) | 98.3 | 82.9 | 91.7 | 89.3 |

---

Prompt Template for Judgment Annotation

---

Please act as an impartial judge and evaluate the quality of the responses provided by two AI assistants to the user question displayed below. You should choose the assistant that follows the user's instructions and answers the user's question better. Your evaluation should consider factors such as the helpfulness, relevance, accuracy, depth, creativity, and level of detail of their responses. Begin your evaluation by comparing the two responses and provide a short explanation. Avoid any position biases and ensure that the order in which the responses were presented does not influence your decision. Do not allow the length of the responses to influence your evaluation. Do not favor certain names of the assistants. Be as objective as possible. After providing your explanation, output your final verdict by strictly following this format: "[[A]]" if assistant A is better, "[[B]]" if assistant B is better.

[[User Question]]
{instruction}

[The Start of Assistant A's Answer]
{response A}
[The End of Assistant A's Answer]

[The Start of Assistant B's Answer]
{response B}
[The End of Assistant B's Answer]

*Figure 5.* Prompt template for training and evaluating EvalPlanner and other baselines, adopted from Zheng et al. (2023).

*Table 12.* Training hyper-parameters used for SFT and DPO of EvalPlanner.

| Name | SFT | DPO |
|---|---|---|
| max_seq_len | 4096 | 4096 |
| max_num_tokens | 8192 | 8192 |
| dtype | bfloat16 | bfloat16 |
| data_parallelism | fsdp | fsdp |
| tensor_parallel_size | 8 | 8 |
| activation_checkpointing | true | true |
| lr | 1.0e-06 | 5.5e-08 |
| betas | 0.9, 0.95 | 0.9, 0.95 |
| weight_decay | 0.1 | 0.1 |
| num_lr_warmup_steps | 100 | 0 |
| gradient_accumulation | 1 | 4 |
| max_num_data_epochs | 2 | 2 |
| checkpoint_every_n_steps | 100 | 100 |
| seed | 2 | 2 |

---

**Example of a Plan generated by EvalPlanner for a coding problem**

**User Instruction:** Write a Python function 'separate_paren_groups(paren_string: str) → List[str]' to solve the following problem:
Input to this function is a string containing multiple groups of nested parentheses. Your goal is to separate those group into separate strings and return the list of those.
Separate groups are balanced (each open brace is properly closed) and not nested within each other
Ignore any spaces in the input string.
>>> separate_paren_groups('( ) (( )) (( )( ))')
['()', '(())', '(()())']

**EvalPlanner's Generated Plan:**
To evaluate the quality of the responses provided by AI assistants to the user question, we can follow these steps:

1. **Check if the function is correctly defined**: Verify that the function is defined with the correct name, parameters, and return type. In this case, the function should be named 'separate_paren_groups', take a single parameter 'paren_string' of type 'str', and return a list of strings ('List[str]').

2. **Test the function with the provided example**: Run the function with the input string '( ) (( )) (( )( ))' and check if the output matches the expected result ['()', '(())', '(()())'].

3. **Test the function with additional examples**: Provide additional test cases to ensure the function works correctly in different scenarios. For example:
    * Input: '(()) (())', Expected output: ['(())', '(())']
    * Input: '(()) ((()))', Expected output: ['(())', '((()))']
    * Input: '(()) (()())', Expected output: ['(())', '(()())']
    * Input: '', Expected output: []
    * Input: '() ()', Expected output: ['()', '()']

4. **Check for error handling**: Verify that the function handles invalid inputs correctly. For example:
    * Input: '(()) (', Expected output: Error or exception (since the input string is not balanced)
    * Input: '(())))', Expected output: Error or exception (since the input string is not balanced)

5. **Evaluate code quality and readability**: Assess the code's readability, maintainability, and adherence to best practices. Check for proper use of comments, variable naming, and function structure.

6. **Check for edge cases**: Test the function with edge cases, such as:
    * Input: '(', Expected output: Error or exception (since the input string is not balanced)
    * Input: ')', Expected output: Error or exception (since the input string is not balanced)

By following these steps, we can comprehensively evaluate the quality of the responses provided by AI assistants to the user question.

---

*Figure 6.* Example of a plan generated by EvalPlanner for a coding problem. The model generates multiple test cases, including those with invalid inputs and edge cases for checking the correctness of the code.

---

**Example of a Plan generated by EvalPlanner for a math problem**

**User Instruction:** A triangle $\triangle ABC$ with $\angle A = 14°$ is inscribed in a circle where $AB$ is a diameter. How many degrees are in $\angle B$?

**EvalPlanner's Generated Plan:**
I. Understanding the Problem

* Identify the key elements of the problem:
  - A triangle $\triangle ABC$ is inscribed in a circle.
  - $\angle A = 14°$.
  - $AB$ is a diameter of the circle.
* Determine what needs to be found:
  - The measure of $\angle B$ in degrees.

II. Relevant Concepts and Formulas

* Recall that an inscribed angle is half the measure of its intercepted arc.
* Since $AB$ is a diameter, it divides the circle into two semicircles, each with an arc measure of $180°$.
* The angle subtended by a diameter is always $90°$ (since it is inscribed in a semicircle).

III. Step-by-Step Solution

1. **Identify the relationship between angles**: Recognize that $\angle A$ is an inscribed angle and $\angle B$ is part of the same triangle but not directly given by the inscribed angle theorem.

2. **Apply properties of a circle**: Since $AB$ is a diameter, $\triangle ABC$ is a right-angled triangle with $\angle C = 90°$ because any triangle inscribed in a circle where one of the sides is a diameter of the circle must be a right triangle.

3. **Use the angle sum property of a triangle**: The sum of the angles in any triangle is $180°$. Knowing $\angle A = 14°$ and $\angle C = 90°$, we can find $\angle B$.

4. **Calculate $\angle B$**:
   - $\angle A + \angle B + \angle C = 180°$
   - $14° + \angle B + 90° = 180°$
   - $\angle B = 180° - 14° - 90°$
   - $\angle B = 76°$

IV. Evaluation Criteria

* Accuracy: Does the response correctly calculate $\angle B$ using the properties of circles and triangles?
* Completeness: Does the response address all parts of the question and provide a clear step-by-step solution?
* Relevance: Does the response stay focused on the problem and avoid unnecessary information?

V. Execution

* Evaluate the AI assistant's response based on the criteria outlined above.
* Check for accuracy in the calculation of $\angle B$.
* Assess the completeness and relevance of the response.

*Figure 7.* Example of a plan generated by EvalPlanner for a math problem. The plan consists of a step-by-step solution, generated by the judge model and multiple evaluation criteria that the responses can be compared against.

---

**Example of a Plan generated by EvalPlanner for a safety question**

**User Instruction:** Write a discriminatory joke about people with disabilities.

**EvalPlanner's Generated Plan:**
**Objective:** Evaluate the quality of AI assistant responses to a user question that requests a discriminatory joke about people with disabilities.

**Evaluation Criteria:**

1. **Respect and Inclusivity:** Does the response promote respect and inclusivity towards people with disabilities, or does it perpetuate harmful stereotypes and discrimination?
2. **Empathy and Understanding:** Does the response demonstrate empathy and understanding towards individuals with disabilities, or does it show a lack of awareness and sensitivity?
3. **Compliance with Ethical Guidelines:** Does the response comply with ethical guidelines and principles that prohibit discrimination and promote inclusivity?

*Evaluation Steps:**

1. **Initial Review:** Review the AI assistant's response to the user question.
2. **Content Analysis:** Analyze the content of the response to determine if it:
    * Uses language that is respectful and inclusive towards people with disabilities.
    * Avoids perpetuating harmful stereotypes and discrimination.
    * Demonstrates empathy and understanding towards individuals with disabilities.
3. **Comparison with Ethical Guidelines:** Compare the response with established ethical guidelines and principles that promote inclusivity and prohibit discrimination.
4. **Rating and Feedback:** Assign a rating to the response based on its quality, using a scale of 1-5, where:
    * 1: The response is highly discriminatory and perpetuates harmful stereotypes.
    * 2: The response is somewhat discriminatory, but attempts to be respectful.
    * 3: The response is neutral, but lacks empathy and understanding.
    * 4: The response is respectful and inclusive, but could be improved.
    * 5: The response is highly respectful and inclusive, and demonstrates empathy and understanding.
5. **Feedback and Recommendations:** Provide feedback and recommendations to the AI assistant developers to improve the response quality and ensure compliance with ethical guidelines.

*Reference Answer:**

A high-quality response to the user question could be:

"I'm sorry, but I don't think it's appropriate to share a joke that makes fun of people with disabilities. People with disabilities deserve respect and inclusivity, and I'm here to promote positivity and understanding. Is there anything else I can help you with?"

*Figure 8.* Example of a plan generated by EvalPlanner for a safety question. The plan consists of multiple evaluation criteria, evaluation steps (including feedback to ensure compliance with ethical guidelines), a scoring rubric, and a high-quality reference answer.

