# OpenReview forum: "Learning to Plan & Reason for Evaluation with Thinking-LLM-as-a-Judge"
_ICML.cc/2025/Conference — ICML 2025 poster_

### Official Review · Reviewer_HUzL · 2025-03-09

**Overall Recommendation:** 2

**Summary:**

The paper proposes a new training pipeline for LLM-as-a-judge models, using online preference optimization techniques as well as an agentic workflow that lets the model first output a plan then the detailed execution followed by the verdict. Experiments indicate the superiority of their new approach

**Claims And Evidence:**

Claims are sound and clear

**Essential References Not Discussed:**

N/A

**Experimental Designs Or Analyses:**

Experimental design is valid

**Methods And Evaluation Criteria:**

Method is intuitive and makes sense

**Other Comments Or Suggestions:**

Missing a limitation section in the main paper

**Other Strengths And Weaknesses:**

# Strengths
- relevant and important topic investigated
- Easy to understand and clear paper writing
- intuitive and simple method

# Weaknesses
- it is unclear this manual decomposition of plan/execute is most optimal for models, or only good for the models tested.
- as the previous point, it is never ablated what happens if you simply CoT->verdict and use online preference optimization to optimize. There are no demonstrations of how the plan->execute->verdict pipeline is superior.

**Questions For Authors:**

- Since you are using offline preference optimization, isn't it ok to simply reverse the order of the candidates and edit the final answer accordingly, instead of actually switching the order and regenerating?
- How is your LLM decoded during evaluation? what was the temperature/top-p? If you had temperature > 0, what are the standard deviation statistics of the results shown?
- Did you try using techniques like self-consistency to further boost your results? did you notice any boost at all?

**Relation To Broader Scientific Literature:**

The method proposed can be a good way to further enhance the LLM-as-a-judge ability of current LLM systems

**Theoretical Claims:**

N/A

---

> ### Author Rebuttal · Authors · 2025-04-01
>
> Thank you for your review!
>
> > it is unclear this manual decomposition of plan/execute is most optimal for models
>
> We added additional experiments to showcase the effectiveness of EvalPlanner (planning+execution) for smaller models. In particular, we experimented with Llama-3.1-8B-Instruct and obtained up to 14 points of absolute improvements -- see our answer to Reviewer 1bZh on “Does this method rely heavily on a strong seed model”.
>
> > it is never ablated what happens if you simply CoT->verdict and use online preference optimization to optimize
>
> Note that we already have such experiments in the paper. First, one of our baselines, Self-Taught Evaluators (Wang et al., 2024) uses similar data to EvalPlanner and performs preference optimization of simpler CoTs without extensive planning. EvalPlanner outperforms it on RewardBench by up to 4 absolute points (90.0 -> 93.9; Table 2), and by 9-10 points on the other benchmarks (Tables 3, 4, 5) . Second, in Table 7, we also ablate the effectiveness of EvalPlanner’s unconstrained plans with other kinds of plans, wherein we show that our plans outperform other baselines like “list of evaluation criteria”.
>
> > isn't it ok to simply reverse the order of the candidates and edit the final answer accordingly
>
> Note that the CoTs (plan+execution) change based on the order of the responses. So, only editing the final answer will make it inconsistent with the corresponding CoT and hence, it is important to perform preference optimization on the data (CoT+final verdict) obtained by reversing the response order.
>
> > How is your LLM decoded during evaluation?
>
> As noted in Line 307, we perform greedy decoding  when inference.
>
> > If you had temperature > 0, what are the standard deviation statistics of the results shown?
>
> We sampled 8 generations with a temperature of 0.8 and top_p of 0.95. Our results on RewardBench are 93.4 with a standard deviation of 0.3. These results are comparable to what we report in our paper, thus showing the effectiveness of EvalPlanner under different decoding hyperparameters.
>
> > Did you try using techniques like self-consistency to further boost your results?
>
> We tried it with 8 samples (using the same temperature and top_p as above) and obtained a score of 93.8. So, we did not observe any further improvement in results. This is expected given the fact that the answer space is limited to only two choices (A or B) and the small standard deviation in our results, as noted in the previous answer.
>
> > Missing a limitation section
>
> We will add this in the next version.

---

### Official Review · Reviewer_dtee · 2025-03-11

**Overall Recommendation:** 3

**Summary:**

This paper introduces EvalPlanner, which is a preference optimization algorithm for thinking-llm-as-a-judge. It first generates an unconstrained evaluation plan, followed by its execution, and then the final judgement. It uses a selftraining loop to iteratively optimizes data and evaluation predictions. The paper conducts extensive experiments to demonstrate the effectiveness.

**Claims And Evidence:**

Based on my review, I did not find any issues with the claims made in the submission.

**Essential References Not Discussed:**

I think most of the key related works are included in the submission

**Experimental Designs Or Analyses:**

The paper conducts extensive experiments to validate the effectiveness on 2 llama-70b LLMs. However, a potential weakness is that it remains unclear whether the proposed method would also be effective on LLMs from other families or smaller-sized models

**Methods And Evaluation Criteria:**

They make sense for me

**Other Comments Or Suggestions:**

see the strengths and weaknesses section

**Other Strengths And Weaknesses:**

strengths:

1. The paper introduces a novel LLM-as-Judge training data synthesis method, which first generates an evaluation plan and then produces step-by-step CoT rationales for each plan.

2. Extensive experiments demonstrate the effectiveness of the proposed method.

weaknesses:

1. The method evaluates only on two llama-70b models. it's unclear whether the proposed method would also be effective on LLMs from other families or smaller-sized models

**Questions For Authors:**

While effective, the EvalPlan model is expected to generate longer responses for evaluation. Compared to other methods, your inference cost is likely to increase—by how much exactly?

**Relation To Broader Scientific Literature:**

Effective reward modeling is crucial. Unlike traditional models that output scalar scores, LLM-as-a-Judge utilizes test-time compute to generate CoT rationales, refining evaluation. EvalPlan tackles the current challenges in collecting high-quality training data, and the resulting model has effective prediction performance.

**Theoretical Claims:**

Does not apply as this paper does not involve any theoretical claims.

---

> ### Author Rebuttal · Authors · 2025-04-01
>
> Thank you for your positive comments about the novelty and the extensive experiments of our paper.
>
> > it remains unclear whether the proposed method would also be effective on LLMs from other families or smaller-sized models
>
> EvalPlanner, does in fact, work well even with smaller sized models. To show this, we conducted additional experiments with Llama-3.1-8B-Instruct – see our answer to Reviewer 1bZh on “Does this method rely heavily on a strong seed model”.
>
> > Compared to other methods, your inference cost is likely to increase—by how much exactly?
>
> EvalPlanner, on average, generates 1K tokens during inference. Note that the goal of EvalPlanner is to indeed increase test-time compute for evaluation. Complex evaluation is a reasoning problem and in line with recent literature on reasoning, we show that evaluation also benefits from expending more test-time compute.

---

### Official Review · Reviewer_1bZh · 2025-03-14

**Overall Recommendation:** 4

**Summary:**

The paper introduces EvalPlanner, a novel preference optimization algorithm designed to enhance the Thinking-LLM-as-a-Judge framework for evaluating LLM responses. The approach employs a self-training loop that iteratively optimizes synthetic evaluation plans and executions using Direct Preference Optimization (DPO). Key algorithmic innovations include generating diverse plans and executions for each instruction-response pair, followed by preference tuning to refine Chain-of-Thought (CoT) reasoning. The experiments demonstrate EvalPlanner’s superior performance over existing methods on benchmarks like RewardBench, RM-Bench, JudgeBench, and Follow BenchEval. The paper also highlights the method’s data efficiency, achieving competitive results with as few as 5K synthetic preference pairs, and its ability to generalize across diverse evaluation tasks, such as coding, math, and safety-related prompts.

## update after rebuttal
Thank you very much for the author's hard work in rebuttal. These replies solved some of my questions. I will raise my rating from 3 to 4. This is because I am paying increasing attention to the evaluation of LLM replies, and this paper makes a pioneering attempt.

**Claims And Evidence:**

The claims in the paper are robustly supported by experimental evidence.

- Through iterative DPO, EvalPlanner demonstrates superior performance across a variety of evaluation tasks.
- The use of an unconstrained evaluation plan enhances its general-purpose planning capability across diverse domains.
- The method’s data efficiency is well-documented, achieving competitive results with as few as 5K synthetically generated preference pairs

**Essential References Not Discussed:**

N/A

**Experimental Designs Or Analyses:**

The experiments in this paper are fairly solid and effectively demonstrate the superiority of the proposed method. However, further analysis in the following directions would enhance the study:

- The paper employs different seed models. It would be helpful if the authors could further discuss the impact of these model choices on the experimental results. I noticed that the performance improvement from **LLaMA 3.3-70B-Instruct** over **LLaMA 3.1-70B-Instruct** is quite significant. Does this method rely heavily on a strong seed model?
- The paper decouples *thoughts* into planning and reasoning. I am curious whether this decoupling contributes to the model’s effectiveness. What would the performance be if only planning or reasoning were used independently?
- The performance improvement with the second iteration is promising. What about further iterations? I appreciate that the authors have already mentioned this in the paper, and I look forward to seeing future results.

I understand that running these additional experiments within the rebuttal phase is challenging, so I encourage the authors to address my questions to the best of their ability. I will take the time constraints into account.

**Methods And Evaluation Criteria:**

The proposed EvalPlanner method and its evaluation criteria effectively assess the model's effectiveness, particularly through pairwise response comparisons. I agree that, compared to scalar-scoring reward models, the LLM-as-a-Judge approach offers greater robustness and interpretability. However, real-world scenarios often require evaluating a single response’s correctness (e.g., determining when self-reflective reasoning should terminate) or identifying the best among multiple sampled responses (e.g., in Tree of Thoughts at each step). Extending the evaluation to these cases could broaden the method’s applicability and practical impact.

**Other Comments Or Suggestions:**

N/A

**Other Strengths And Weaknesses:**

**Strengths:**

The paper presents several strengths for EvalPlanner:

1. It achieves performance surpassing baselines with fewer prompt pairs, demonstrating data efficiency with as few as 5K synthetic preference pairs.
2. It eliminates the need for human-annotated data by leveraging synthetically generated CoTs, enhancing scalability.
3. The unconstrained planning approach fosters general-purpose planning across multiple domains, as evidenced by its strong results on diverse benchmarks.
4. Future work could integrate EvalPlanner as a reward model in RLHF workflows, indicating promising adaptability for broader applications.

**Weaknesses:**

1. The method is primarily evaluated on pairwise response comparisons, raising questions about its applicability to broader scenarios, such as ranking multiple responses/assess single response. While multi-response comparison could be an extension of pairwise evaluation (potentially achievable with this approach), its effectiveness in such cases remains untested.
2. The integration of EvalPlanner with a wider range of CoT-based methods is underexplored, limiting insights into its compatibility with other reasoning frameworks beyond the proposed planning-execution-judgment structure.
3. The paper lacks deeper experimental analysis, such as the impact of the choice of seed model or a more granular ablation on decoupling planning and reasoning, which could clarify their individual contributions to performance.
4. Can you give the training cost of SFT and DPO?

Detailed comments are provided above.

**Questions For Authors:**

I would greatly appreciate it if the authors could address my questions and clarify the broader application scope of this method. In that case, I would consider adjusting my rating accordingly.

**Relation To Broader Scientific Literature:**

- This method could be integrated with variations of Chain-of-Thought (CoT), many of which require evaluation of responses, such as **Tree of Thoughts, Graph of Thoughts, self-reflection, and LLM blender**. It could serve as a general evaluator within these frameworks.
- Additionally, some methods analyze responses (mostly CoT-based) at a finer granularity, assessing the information gain and correctness of each step. In the context of long cot reasoning, this method could potentially be extended to evaluate each step individually.

**Theoretical Claims:**

This paper primarily relies on empirical evidence to demonstrate the effectiveness of the proposed method, while lacking theoretical support. This is understandable, and I do not insist that the authors provide additional theoretical claims.

---

> ### Author Rebuttal · Authors · 2025-04-01
>
> Thank you for your review and appreciating our work! We are also glad to hear that you’re willing to adjust your scores. We respond to your comments below.
>
> > Evalplanner’s applicability to Best-of-N settings
>
> Upon your suggestion, we conducted some experiments and obtained promising results. Please refer to our response to Reviewer bPFz on “Assess whether the proposed judge can be effectively applied to rejection sampling”
>
> > Does this method rely heavily on a strong seed model
>
> No, EvalPlanner works equally well, even with 8B models. We happened to experiment with a stronger seed model to maximize performance. As shown below, EvalPlanner w/ Llama-3.1-8B-Instruct improves the seed model by a large 14 points (69.5 → 83.0), almost matching the performance of the much larger Llama-3.1-70B-Instruct and Claude-3.5-Sonnet. Evalplanner does not make any model-specific assumptions and hence should be expected to work with any model for scaling up test-time compute for evaluation.
>
> |                                            | **Overall** | **Chat** | **Chat-Hard** | **Safety** | **Reasoning** |
> |--------------------------------------------|-------------|----------|---------------|------------|---------------|
> | **Llama 3.1-8B Instruct (seed)**           |     69.5    |   92.7   |      46.1     |    64.4    |      74.7     |
> | **Llama 3.1-70B Instruct (seed)**          |     84.1    |   97.2   |      70.2     |    82.8    |      86.0     |
> | **Claude-3.5-Sonnet**                      |     84.2    |   96.4   |      74.0     |    81.6    |      84.7     |
> | **EvalPlanner (w/ Llama-3.1-8B-Instruct)** |     83.0    |   85.5   |      84.0     |    83.4    |      79.3     |
>
> > What would the performance be if only planning or reasoning were used independently
>
> Note that we do such ablations in the paper. First, one of our baselines –- Self-Taught Evaluators (Wang et al., 2024) — trains a judge that generates CoTs without a planning component. Next, in Table 7, we compare EvalPlanner’s plans to other kinds of constrained plans. Having an explicit step-by-step plan allows the model to better reason through it, leading to much better performance. Since the reasoning component always has to be there to produce a verdict, our paper contains ablations of (1) no plan, and (2) other kinds of plans. We’ll clarify this more in a future version.
>
> > The performance improvement with the second iteration is promising. What about further iterations?
>
> We did not try a third iteration because of the computational overhead associated with online DPO. This requires obtaining newer/harder prompts and preference pairs, generating outputs from the previous iteration of model, preparing data, and performing preference optimization. That said, with the right kind of data, we believe further iterations should lead to more improvements. We hope that future work can explore it in more detail.
>
> > The integration of EvalPlanner with a wider range of CoT-based methods is underexplored
>
> We already explored different forms of CoTs in the paper in Table 7 by varying the type of plan that the judge generates.
>
> > The paper lacks deeper experimental analysis, such as the impact of the choice of seed model
>
> Refer to our experiments with a Llama 8B model.
>
> > a more granular ablation on decoupling planning and reasoning
>
> We already have such ablations in our paper. First, one of our baselines, Self-Taught Evaluators (Wang et al., 2024) uses similar data to EvalPlanner and performs preference optimization of simpler CoTs without planning. So, it should be seen as a baseline with no planning. EvalPlanner outperforms it on RewardBench by up to 4 absolute points (90.0 --> 93.9; Table 2), and by 9-10 points on the other benchmarks (Tables 3, 4, 5) . Second, in Table 7, we also ablate the effectiveness of EvalPlanner’s unconstrained plans with other kinds of plans, wherein we show that our plans outperform other baselines like “list of evaluation criteria”.
>
> > Can you give the training cost of SFT and DPO?
>
> All our experiments are performed on A100 GPUs. SFT of a Llama 70B model requires 3 nodes (24 A100 GPUs) and DPO requires 8 nodes (64 A100 GPUs).
>
> > clarify the broader application scope of this method
>
> Evaluation, as we hypothesize in the paper, is a reasoning problem where the judge should plan for the evaluation recipe and then reason through it to arrive at the verdict. Current literature on reasoning (O1/R1) has shown major improvements by scaling up test-time compute. EvalPlanner should be seen as one of the first SOTA recipes for scaling up test-time compute, specifically for evaluation. Through additional Best-of-N experiments, we have also established the effectiveness of Thinking-LLM-as-a-Judge models like our EvalPlanner in improving policy models. Future work could explore its applicability in RLHF pipelines, where both generation and evaluation are scaled up at test time.

---

### Official Review · Reviewer_bPFz · 2025-03-14

**Overall Recommendation:** 2

**Summary:**

This paper proposes EvalPlanner, a method that separates planning from reasoning to enhance LLM-as-a-Judge evaluation. EvalPlanner iteratively improves itself using synthetic preference pairs, achieving state-of-the-art performance (93.9%) on RewardBench and strong results on RM-Bench, JudgeBench, and FollowBenchEval​

**Claims And Evidence:**

Most of the claims in the paper are well-supported by empirical evidence.

To further strengthen the justification of the proposed method’s effectiveness, the following additional studies would be valuable:

1. Assess whether the proposed judge can be effectively applied to rejection sampling, Direct Preference Optimization (DPO), or online reinforcement learning (RL).
2. Analyze how the model's performance evolves over different stages of iterative training to validate its self-improvement capability.
3. Examine how varying the amount of training data affects performance to determine the model’s data efficiency and scalability.

**Essential References Not Discussed:**

No

**Experimental Designs Or Analyses:**

See above "Claims And Evidence" section.

**Methods And Evaluation Criteria:**

Yes, the proposed method is well motivated and makes sense. It is just a bit straightforward and seems to be an extension of [Pang et al., 2024], [Wang et al., 2024], and [Wu et al., 2024b] by adapting to LLM evaluation tasks with an additional planning step.

The used datasets are comprehensive.

**Other Comments Or Suggestions:**

- It appears that the performance scores for "safety" and "reasoning" have been incorrectly placed for some baselines under the "Reward Models with Critiques" category. Could the authors clarify or correct this?

- Adding 1-2 small experiments demonstrating how the proposed evaluator can enhance generation quality in other policy models would strengthen the practical impact of the method.

**Other Strengths And Weaknesses:**

n/a

**Questions For Authors:**

- The baselines in the paper appear to use different training datasets, making it unclear how fair comparisons are ensured. Could the authors clarify how differences in training data impact the results?

- The experiments are conducted exclusively with LLaMA-3-70B models. Does this suggest that the proposed method relies on a strong base model as a prerequisite for effectiveness?

- If without control, the llm generated planning and evaluation may still contain systematic bias. What is the potential way to address this issue to make sure it makes fair evaluations?

**Relation To Broader Scientific Literature:**

While the proposed training approach exists in existing literature, the application to llm-as-a-judge scenarios seems novel.

**Theoretical Claims:**

N/A

---

> ### Author Rebuttal · Authors · 2025-04-01
>
> Thank you for your review!
>
> > Assess whether the proposed judge can be applied to rejection sampling …
>
> First, note that performing extensive RLHF experiments with Evalplanner is beyond the scope of this work and requires separate studies. That said, upon your suggestion, we conducted additional Best-of-N experiments with Evalplanner, on two hard reasoning benchmarks  – GPQA (diamond) and AIME2024, obtaining promising results. The experimental setup is as follows:
>
> * For each test data point, we sample N (= 8/16) responses from Llama-3.1-70B-Instruct.
> * Since Evalplanner is a pairwise judge, we then prepare samples of response-pairs, amounting to N*(N-1) pairs (considering both orders) for each data point.
> * Then we evaluate all these pairs using EvalPlanner and compute ELO ratings to rank the N responses.
> * As baselines, we also report results for pass@1, random@N, and self-consistency@N. As an upper bound, we report pass@N.
>
> Below are the results on GPQA (Diamond), where BoN with EvalPlanner improves pass@1 by up to 5 absolute points.
>
> |                                    | **N=8**  | **N=16** |
> |------------------------------------|----------|----------|
> | **Pass@1**                         |   42.1   |   42.9   |
> | **Pass@N**                         |   80.8   |   87.3   |
> | **Random@N**                       |   41.6   |   43.4   |
> | **Self-Consistency@N**             |   42.4   |   46.4   |
> | **Best-of-N (w/ EvalPlanner 3.3)** | **44.5** | **47.8** |
>
> Next, on AIME2024, EvalPlanner also improves Pass@1 by up to 15 absolute points.
>
>
> |                                    | **N=8**  | **N=16** |
> |------------------------------------|----------|----------|
> | **Pass@1**                         |   21.6   |   20.8   |
> | **Pass@N**                         |   43.3   |   43.3   |
> | **Random@N**                       |   23.3   |   20.0   |
> | **Self-Consistency@N**             |   20.0   |   20.0   |
> | **Best-of-N (w/ EvalPlanner 3.3)** | **36.7** | **30.0** |
>
>
> Both these results show the promise of EvalPlanner in improving LLMs on downstream tasks.
>
> > Analyze how the model's performance evolves over different stages of iterative training to validate its self-improvement capability.
>
> Note that our paper already contains these results in Tables 2 and 3 (with related discussions in Section 4). We present them again below. Recall that EvalPlanner consists of one iteration of SFT and two iterations of DPO. On RewardBench, the accuracies after these three stages are 86.8, 92.3, and 93.9. SFT doesn’t improve results much, especially on the Chat-hard category which contains subtle differences in response-pairs. After we construct DPO pairs teaching models to recognize these differences, it leads to major improvements in the first iteration, even with a small number of prompts (5K). In the second iteration, we obtain further improvements.
>
> > Examine how varying the amount of training data affects performance to determine the model’s data efficiency and scalability.
>
> Once again, these results were already presented in the paper in Table 2, associated with a section dedicated specifically to "EvalPlanner is data-efficient and benefits from iterative
> thought optimization". We show that with as few as 5K synthetic preference pairs, EvalPlanner is competitive with SOTA reward models on RewardBench, obtaining accuracy of 92.3.
>
> > It is just a bit straightforward…
>
> Straightforward methods that work well in practice should be preferred. In terms of novelty, to our knowledge, we are the first ones to design a SOTA method that leverages test-time compute for evaluation (via planning and reasoning).
>
> > It appears that the performance scores for "safety" and "reasoning" have been incorrectly placed for some baselines
>
> Thanks for noting this! We’ll fix this in the next version.
>
> > how the proposed evaluator can enhance generation quality in other policy models
>
> Refer to the answer to your first question!
>
> > The baselines in the paper appear to use different training datasets, making it unclear how fair comparisons are ensured.
>
> * First, one of the baselines – Self-taught Evaluator (Wang et al., 2024) uses training data similar to ours and we outperform them by a significant margin.
> * Second, for the other baselines, we did not need to match their training data to show the effectiveness of EvalPlanner because EvalPlanner only relies on synthetic pairs and a much smaller number of them. With more or human-annotated data, we expect EvalPlanner to scale even better. This has generally been shown for thinking models that expend more test-time compute for reasoning.
>
> > EvalPlanner with weaker seed models
>
> Refer to our response to Reviewer 1bZh on “Does EvalPlanner rely on a strong seed model”.
>
> > llm generated planning and evaluation may still contain systematic bias..
>
> It’s a possibility but compared to scalar RMs, we expect a model like EvalPlanner that generates CoTs to be better and more interpretable when dealing with biases.

---

> > ### Comment · Reviewer_bPFz · 2025-04-03
> >
> > Thank you for your comments. I have a few additional points:
> >
> > 1. I appreciate the inclusion of best-of-N results. However, I noticed that comparisons with other judge/reward models were not provided, which makes the evaluation a bit less comprehensive.
> >
> > 2. Regarding the bias issue, I'm not entirely convinced by the current argument. For instance, what about the possibility that the chain-of-thought (CoT) approach might be even more misleading or introduce its own biases? It would strengthen the discussion to include some case studies or empirical comparisons to support this point.

---

> > > ### Author Response · Authors · 2025-04-07
> > >
> > > > Comparison with other judge models
> > >
> > >
> > > We added a couple of other baselines for Best-of-N with two different judges – (1) the seed Llama model and (2) Self-Taught Evaluators (Wang et al., 2024) which is also trained on a similar amount of data as EvalPlanner. EvalPlanner outperforms both these baselines on AIME2024 by a significant margin.
> > >
> > > |                                            |  **N=8** |
> > > |--------------------------------------------|:--------:|
> > > | **Pass@1**                                 |   21.6   |
> > > | **Pass@N**                                 |   43.3   |
> > > | **Random@N**                               |   23.3   |
> > > | **Self-Consistency@N**                     |   20.0   |
> > > | **Best-of-N (w/ seed Llama-70B-Instruct)** |   16.7   |
> > > | **Best-of-N (w/ Self-Taught Evaluator)**   |   26.7   |
> > > | **Best-of-N (w/ EvalPlanner 3.3)**         | **36.7** |
> > >
> > > We'd again like to note that in line with most prior works on Reward Modeling (SFR-LLaMA-3.1-70B-Judge, CLoud, Self-Taught Evaluators, Critic-RM, etc) we evaluate EvalPlanner on standard reward modeling benchmarks. These additional BoN experiments should be seen as additional evidence about the effectiveness of EvalPlanner. However, performing extensive alignment experiments with EvalPlanner is beyond the scope of this paper and we hope that future work can build on top of our work.
> > >
> > > > chain-of-thought (CoT) approach might be even more misleading or introduce its own biases
> > >
> > > Recent works suggest that CoT monitoring can be far more effective than monitoring agent actions and outputs alone (see Baker et al., 2025: https://arxiv.org/abs/2503.11926). Regardless, this is a topic that extends much beyond EvalPlanner and requires further research for thinking models, in general, that generate CoTs. We believe such research would benefit Thinking-LLM-as-a-Judge models as well like EvalPlanner.
> > >
> > > We have tried to answer your questions by conducting multiple additional experiments and if they have, we would greatly appreciate it if you could revisit your score accordingly.

---

### Decision · Program_Chairs · 2025-05-01

**Decision:**

Accept (poster)

**Comment:**

The paper proposes EvalPlanner a preference optimization algorithm for Thinking-LLM-as-a-Judge that first generates an unconstrained evaluation plan, followed by its execution, and then the final judgment. In a self-training loop, EvalPlanner iteratively optimizes over synthetically constructed evaluation plans and executions, leading to better final verdicts. The paper focuses on an important and challenging problem in the current SoTA thinking modelsThe claims are supported with empirical study on big Llama (and small llama and a Claude model). Authors sufficiently answered questions raised by the authors. please update the paper with the experiments and clarifications provided during the rebuttal period.